# Detection of c.375A>G, c.385A>T, c.571C>T, and *se^del2^* of *FUT2* via Real-Time PCR in a Single Tube

**DOI:** 10.3390/diagnostics13122022

**Published:** 2023-06-10

**Authors:** Mikiko Soejima, Yoshiro Koda

**Affiliations:** Department of Forensic Medicine, Kurume University School of Medicine, Kurume 830-0011, Japan; misoe@med.kurume-u.ac.jp

**Keywords:** *Alu*-mediated nonhomologous recombination, *FUT2*, *se^del2^*, c.375A>G, c.385A>T, c.571C>T, Polynesian population

## Abstract

α(1,2)fucosyltransferase (Se enzyme) encoded by *FUT2* is involved in the secretor status of ABH(O) blood group antigens. The *se^del2^* allele is one of the non-functional *FUT2* (*se*) alleles in which 9.3 kb, containing the entire coding region of *FUT2*, is deleted by *Alu*-mediated nonhomologous recombination. In addition to this allele, three SNPs of *FUT2*, c.375A>G, c.385A>T, and c.571C>T, appear to be prevalent in certain Oceanian populations such as Polynesians. Recently, we developed an endpoint genotyping assay to determine *se^del2^* zygosity, using a FAM-labeled probe for detection of the *se^del2^* allele and a VIC-labeled probe for the detection of *FUT2*. In this study, instead of the VIC probe, a HEX-labeled probe covering both c.375A>G and c.385A>T and a Cy5-labeled probe covering c.571C>T were added to the *se^del2^* allele assay mixture to allow for the simultaneous detection of these four variations via endpoint genotyping for *se^del2^* zygosity and fluorescence melting curve analysis for c.375A>G, c.385A>T, and c.571C>T genotyping. The results obtained from 24 Samoan subjects using this method were identical to those obtained using previous methods. Therefore, it appears that the present method can accurately determine these four variations simultaneously.

## 1. Introduction

α(1,2)fucosyltransferase (Se enzyme) encoded by *FUT2* is involved in the secretor status of ABH(O) blood group antigens [1,2,3]. Secretors carry at least one functional *FUT2* (*Se*) allele, which encodes the active Se enzyme, and consequently expresses ABH antigens in their secretions. In contrast, individuals with only Se enzyme-deficient (*se* or *Se^w^*) alleles either do not express these antigens (non-secretors) or express them in their secretions weakly (weak secretors) [3]. *FUT2* is located in the region of chromosome 19q13.3, together with a pseudogene (*SEC1P*) with high sequence similarity [4,5].

Several single-nucleotide polymorphisms (SNPs) of *FUT2* have been identified [6]. Among these, the nonsense SNP at c.428G>T (W143X, rs601338) is responsible for Se enzyme inactivation of the *se^428^* allele, which is found at approximately 50% frequency in Europe, Africa, and West Asia [6]. On the other hand, the missense SNP at c.385A>T (I129F, rs1047781) is responsible for partial inactivation of the Se enzyme in a representative *Se^w^* allele [6]. This allele is present in East and Southeast Asians at about 50% frequency, and is also common in certain Oceanian populations such as Polynesians [7]. The missense SNP at c.302 C>T (I101P, rs200157007) constitutes the *se^302^* allele, which is common in South Asian populations [6]. The nonsense SNP at c.571C>T (R191X, rs1800028) constitutes the *se^571^* allele, which is relatively common in Polynesian and Taiwanese populations [7,8,9,10]. The synonymous SNP at c.375A>G (rs1800026) has been found in certain New Guineans (Melanesians) with relatively high frequency (more than 20%) and in Africans and Samoans (Polynesians) with relatively low frequency (about 2%) [7,11]. 

In addition, five *Se* alleles resulting from non-allelic homologous recombination have been identified and shown to have a population-specific distribution [7,12]. The *se^fus^* allele occurs due to recombination between *SEC1P* and *FUT2* and seems to be specific to the Japanese population [13], whereas the *se^del^*, *se^del2^*, *se^del3^*, and *se^del4^* alleles are deletions of the entire coding region of *FUT2*, resulting from the recombination of two interspersed repeat elements [7,12]. The *se^del^
*allele is a 10 kb deletion that occurs due to recombination between two *Alu* elements and appears to be characteristic of South Asian populations, with frequencies ranging from 10 to 20%, while the *se^del2^* allele is a 9.3 kb deletion that occurs due to recombination between two *Alu* elements, different from the *se^del^* allele, and appears to be characteristic of certain Oceanian populations such as Samoans (Polynesians) and New Guineans (Melanesians), with frequencies ranging from 10 to 20% [7,11]. The *Alu* elements are the most abundant repeat elements, each approximately 300 bp in length, and occupy about 10% of the human genome. Based on their abundance and sequence similarity, the *Alu* elements are thought to be involved in genomic rearrangements in the human genome [14,15,16]. The *se^del3^
*and *se^del4^* alleles are very rare, currently found in only one Chinese and one Peruvian [12].

TaqMan assays using dual-labeled fluorescence oligonucleotide probes (TaqMan probes) allow for real-time PCR monitoring for DNA and RNA quantification and the detection of SNPs [17]. This is because when the probe hybridizes to the complementary target DNA, it is cleaved by the 5’-3’ exonuclease activity of *Taq* polymerase, causing the quencher to separate from the fluorophore and emit fluorescence [17]. Recently, we developed an endpoint genotyping assay using two TaqMan probes, which were a FAM-labeled probe for detection of the *se^del2^* allele and a VIC-labeled probe for detection of the *FUT2* coding region [12]. The TaqMan probes are also available for fluorescence melting curve analysis (FMCA) to detect SNPs [18,19,20,21]. FMCA is one of the most robust SNP detection methods [18]. Unlike the TaqMan assay, it would be desirable for FMCA to have no degradation of the fluorescent probe. However, it has been reported that *Taq* polymerases both with and without 5’-3’ exonuclease activity are suitable for FMCA under asymmetric PCR conditions, probably because suppressing primer extension of the sense strand of the probe inhibits probe hydrolysis [19]. 

In this study, we attempted to develop a multiplex FMCA for the purpose of developing a genetic analysis method that enables simple and rapid estimation of the secretor status of Oceanians. Since HEX is the most widely used fluorophore that can be used in the same filters as VIC, HEX probes were used instead of VIC. Therefore, a HEX-labeled probe covering c.375A>G and c.385A>T and a Cy5-labeled probe covering c.571C>T were added to the assay mixture of the FAM-labeled probe for detection of the *se^del2^* allele, allowing for the determination of *se^del2^* zygosity via endpoint genotyping, and three SNPs of *FUT2* via FMCA could be identified in a single tube. 

## 2. Materials and Methods

The research protocol was reviewed and approved by the ethical committee of Kurume University School of Medicine (approval No. 22158). 

### 2.1. DNA Samples

Genomic DNA from 24 Samoans in Apia was used. Their *se^del2^* zygosity and all SNPs in the *FUT2* coding region were previously determined via conventional PCR for amplification of the 2.6 kb junction region of the *se^del2^
*allele and the direct Sanger sequencing of PCR products [7]. 

### 2.2. Probes and Primers Used in This Study

The nucleotide position numbers for *FUT2* and *SEC1P* follow those reported by Kelly et al. [4]. Because *FUT2* and *SEC1*P have high DNA sequence similarity, we selected primers of *FUT2* that would not amplify *SEC1* [4]. 

For detection of the *se^del2^* allele, we used the FAM-labeled hydrolysis probe (sedel2-probe; 5′-FAM-CCAGTCTGGCCAACAT-MGB-3′) and a set of primers (sedel2-F primer; 5′-CCGCAATAGAAAGACGTGGA-3′ and sedel2-R primer; 5′-CCAGGTTCAAGCGATTCTTC-3′) that were the same as those described previously and are indicated in Figure 1 A and B [12]. The primers and probes for the detection of c.375A>G, c.385A>T, and c.571C>T are indicated in Table 1 and Figure 1C. Primers for amplification of the *FUT2* fragment surrounding c.571C>T were designed using Primer3Plus (https://www.bioinformatics.nl/cgi-bin/primer3plus/primer3plus.cgi, accessed on 1 March 2023) [22]. Either a HEX-labeled 385-probe—identical to 411–435 bp of *SEC1P* and 369–393 bp of the 385A allele of *FUT2* but one base different from the 385T allele and 375G allele (wild-type, 385A allele with 375G substitution)—or a HEX-labeled 375-probe—identical to the 411–435 bp of *SEC1P* and 369–393 bp of the 375G allele of *FUT2,* but one base different from the wild-type, 385A allele and two bases different from the 385T allele—was used for the detection of c.375A>G and c.385A>T. In addition, a Cy5-labeled FUT2-571C probe was used for the detection of c.571C>T. We also compared FMCA using a 281-bp PCR amplicon from 337 to 617 bp of *FUT2* and 76-bp PCR amplicon from 337 to 412 bp of *FUT2*, plus a 79 bp amplicon from 539 to 617 bp of *FUT2*.

### 2.3. Probe Selection for Detection of c.375A>G and c.385A>T by FMCA

To select probes for the detection of c.375A>G and c.385A>T, asymmetric real-time PCR and FMCA were performed with the following reaction mixture (10 μL total): 5 ng genomic DNA, 5 µL of Premix Ex Taq™ (Probe qPCR) (Takara, Tokyo, Japan) containing *Taq* polymerases with 5’-3’ exonuclease activity, 50 nM FUT2-337F primer, 500 nM FUT2-412R primer, and 200 nM HEX-labeled 385-probe or HEX-labeled 375-probe. The FUT2-337F and FUT2-412R primers were identical to those used for 385A>T detection via unlabeled probe HRM analysis [23]. 

### 2.4. Evaluation of the Number of Amplicons in Genotyping of c.375A>G, c.385A>T, and c.571C>T of FUT2 via FMCA

To compare one amplicon and two amplicons in the genotyping of c.375A>G, c.385A>T, and c.571C>T of *FUT2* via FMCA, asymmetric real-time PCR and FMCA were performed with the following reaction mixture (10 μL total): 5 ng genomic DNA, 5 µL of Premix Ex Taq (Probe qPCR), 200 nM HEX-labeled 375-probe, and 200 nM Cy5-labeled 571 probe. In addition, 50 nM FUT2-337F primer and 500 nM FUT2-617R primer were included for the 281 bp amplicon. Alternatively, 50 nM FUT2-337F primer and 500 nM FUT2-412R primer for the 76 bp amplicon and 50 nM FUT2-531F primer and FUT2-617R primer for the 79 bp amplicon were included.

### 2.5. Detection of c.375A>G, c.385A>T, and c.571C>T via FMCA and Detection of se^del2^ via Endpoint Genotyping in a Single Tube

To detect c.375A>G, c.385A>T, c.571C>T, and the *se^del2^
*allele in a single tube, real-time PCR, FMCA, and endpoint genotyping were performed with the following reaction mixture (10 μL total): 5 ng genomic DNA, 5 µL of Premix Ex Taq (Probe qPCR), 50 nM each of sedel2-F and sedel2-R primers, and 100 nM of sedel2-probe; 50 nM FUT2-337F primer and 500 nM FUT2-617R primer; and 200 nM HEX-labeled 375-probe and 200 nM Cy5-labeled 571 probe. 

### 2.6. Real-Time PCR Monitoring, FMCA, and Endpoint Genotyping 

In this study, real-time PCR monitoring, FMCA, and endpoint genotyping were all performed using the LightCycler 480 instrument II (Roche Diagnostics, Tokyo, Japan). The thermal conditions for all PCRs were identical: preheating at 95 °C for 30 s, followed by 45 cycles of denaturation at 95 °C for 5 s, and annealing/extension at 63 °C for 15 s. The fluorescence data were collected at the end of the annealing/extension step of each cycle using a FAM (465 nm excitation and 510 nm emission), VIC/HEX/Yellow 555 (533 nm excitation and 580 nm emission), and/or Cy5/Cy5.5 (618 nm excitation and 660 nm emission) filters. 

For melting curve genotyping, PCR products were heated to 95 °C for 1 min and rapidly cooled to 40 °C for 1 min, and then, the fluorescence data were collected over a range of 50 to 80 °C, increasing at 0.10 °C/s with 2 to 6 acquisitions/s using the same filters. Endpoint genotyping, melting curve genotyping, and melting temperature (Tm) calculation were performed using LightCycler 480 gene scanning software with default settings.

## 3. Results

### 3.1. Probe Selection for Genotyping of c.375A>G and c.385A>T via FMCA

As mentioned above, in Oceanian populations, including Samoans, a synonymous SNP at c.375A>G of *FUT2* has been observed. In fact, one of our Samoan subjects was heterozygous for c.375A>G (375A/G), and the remaining twenty-three Samoan samples were all homozygous for 375A (375A/A) [7]. The 375A allele is further segregated into 385A and 385T alleles at c.385A>T; the nucleotide at position 385 of the 375A allele is an A, indicating that the 375A/G subjects are heterozygous for the 375G and 385A alleles. We first examined whether c.385A>T and c.375A>G could be separated using a 25 bp hydrolysis (TaqMan) probe, 385-probe, for the same sequence as the 385A (wild-type) allele used previously [24]. As shown in Figure 2A, the Tm value of the 385A allele was around 72 °C and clearly separated from the 385T allele and the 375G allele; however, the Tm values of the 385T and 375G alleles were both around 67 °C when using this probe. Although the peak height of the 375G allele seems to be slightly lower than that of the 385T allele, accurate separation is considered difficult. Therefore, we next examined the 375-probe, which has the same sequence as the 375G allele of *FUT2* and *SEC1P* previously used for the simultaneous detection of 385A>T and the *se^fus^* allele [13]. The 385A allele peak with a Tm around 68 °C could be separated from the 375G allele peak with a Tm around 74 °C and the 385T allele peak with a Tm around 62 °C (Figure 2B). Although the peak with a Tm around 74 °C might also be generated if a corresponding region of the SEC1P was amplified, the primers used in this study amplified only FUT2 and not SEC1P, allowing us to examine the presence of the 375G allele and separate it from the 385A and 385T alleles. Therefore, in this study, we used the 375-probe instead of the 385-probe for the detection of c.375A>G and c.385A>T.

### 3.2. Evaluation of the Number of Amplicons in Genotyping of c.375A>G, c.385A>T, and c.571C>T of FUT2 via FMCA

We then compared the FMCA results using a 281 bp PCR amplicon from 337 to 617 bp of *FUT2* (one amplicon) and a 76 bp PCR amplicon from 337 to 412 bp of *FUT2*, plus a 79 bp amplicon from 539 to 617 bp of *FUT2* (two amplicons). The primers used for PCR amplification are listed in Table 1. As shown in Figure 3A, we were able to separate the 571C allele signal with a Tm around 72 °C from the 571T allele signal with a Tm around 64 °C, and the peak heights of the 571-probe obtained with two amplicons and those with one amplicon are almost equivalent. Although none of the 24 Samoans were 571T/T subjects, one 571T hemizygote (571T/*se^del2^* subject) showed only one melting peak corresponding to 571T. The melting peak pattern of this subject is considered identical to that of the 571T/T subject. Therefore, it would be possible to detect the 571T/T subject. On the other hand, the peaks of the 375-probe obtained with the two amplicons were higher than those obtained with one amplicon, but smaller peaks were observed with the two amplicons that were almost identical to the Tm value of the 375G allele. In addition, as shown in Figure 1, the FUT2-412R primer for two PCR amplicons contained a c.400G>A SNP (V134I, rs370886251), which is relatively common in Melanesians such as New Guineans, but is rarely present in peripheral populations such as Polynesians and almost always absent in other populations [7,11]. In fact, one of the 24 Samoans was heterozygous for 400G/A (G is on 385T allele and A is on 385A allele), and the peak height corresponding to 385A was quite low in this subject when using two amplicons in FMCA (Figure 3C), probably due to a single base mismatch in the FUT2-412R primer. Therefore, there would be a risk of misdiagnosis with a homozygote 385T allele in this subject. For these reasons, we decided to perform FMCA with one amplicon instead of two amplicons in this study.

### 3.3. Genotyping of c.375A>G, c.385A>T, and c.571C>T of FUT2 via FMCA and Endpoint Genotyping of the se^del2^ Allele in a Single Tube

Finally, we evaluated a simultaneous assay of FMCA genotyping of c.375A>G, c.385A>T, and c.571C>T of *FUT2* and endpoint genotyping of the *se^del2^
*allele using 24 Samoans whose *se^del2^* zygosity and all SNPs in the *FUT2* coding region had been previously determined via direct sequencing of PCR products using Sanger sequencing [7]. The representative Sanger sequencing results for the wild-type homozygote, heterozygote, and variant-type homozygote of c.385A>T and c.571C>T, respectively, are shown in Appendix A. As mentioned above, we clearly identified the 375G, 385A, and 385T alleles using the VIC/HEX/Yellow 555 filter (Figure 4A), and the 571C and 571T alleles using the Cy5/Cy5.5 filter (Figure 4B). 

Endpoint genotyping was then performed using the same primer sets previously reported for detection of the *se^del2^
*allele and the HEX-labeled 375 probe or Cy5-labeled 571 probe, instead of the FUT2-specific VIC probe, for detection of the *FUT2* signal [12]. In the present assay, the *se^del2^* allele was detected using a FAM-labeled probe, and the *FUT2* alleles were detected using a HEX-labeled probe or a Cy5-labeled probe. This method completely discriminated the three genotypes, one homozygote of the *se^del2^
*allele (only FAM signal), three heterozygotes of the *se^del2^
*allele (with FAM, HEX, and Cy5 signals), and 20 subjects without the *se^del2^
*allele (without FAM signal and with HEX and Cy5 signals) for 24 Samoans. Scatter plots of endpoint genotyping are shown in Figure 5A,B. The subjects with 385A/- (385A/*se^del2^*) were separated from those with 385T/- (385T/*se^del2^*) in a dual-color scatter plot of fluorescence signals because the HEX/FAM signal ratios of 385A/*se^del2^* were approximately 3 times higher than those of 385T/*se^del2^* (Figure 5A). This may be because the 375-probe dissociates mostly from the T allele, whereas it binds completely to the A allele at 63 °C, which is the temperature of annealing/extension that collects the real-time PCR fluorescence signal. On the other hand, no such differences were observed between the subjects with 571C/*se^del2^* and 571T/*se^del2^* (Figure 5B). During all events, the results of the *se^del2^* zygosity determined using this method were also in perfect agreement with those determined using previous genotyping methods [7].

## 4. Discussion

Recently, we developed endpoint genotyping to determine *se^del2^* zygosity using primers and a FAM-labeled sedel2-probe for the amplification and detection of *se^del2^* and primers, and a VIC-labeled probe for *FUT2* [12]. In this study, we applied this method to the determination of three SNPs of *FUT2* (c.375A>G, c.385A>T, and c.571C>T), as well as the determination of *se^del2^* zygosity. The present method appears to be advantageous in terms of cost and time savings compared to the previous method that determined only the *se^del2^* zygosity, because the present method could estimate the secretor status of Polynesian subjects in a single assay [12]. The direct Sanger sequencing of PCR products is the gold standard method for the detection and discovery of SNPs [25]. However, its disadvantage is that it requires rather complicated post-PCR operation. On the other hand, FMCA is a robust and simple method for the detection of known SNPs, is capable of multiplex assays, and is suitable for relatively large-scale analysis, but it is not suitable for the detection of unknown SNPs [19]. Another real-time PCR-based method that does not use labeled probes, high-resolution melting analysis, is faster, simple, and particularly suitable for detecting heterozygotes, making it an excellent method for determining infrequent SNPs, and can also detect unknown SNPs within the amplified region [26]. The disadvantage of this method, however, is that the difference in Tm between wild-type and variant-type homozygotes is often small, within 1 °C, making it unsuitable for separating each homozygote. Therefore, it is necessary to use these methods differently depending on the distribution of variations. The limitations of this method are that it cannot detect Se alleles other than *Se^w^*, *se^57^*^1^, and *se^del2^*, and it is not very useful for *FUT2* polymorphism analysis outside of Oceania.

Given the characteristic distribution of *Se* alleles due to non-allelic homologous recombination events, it is likely that the *se^fus^* allele was generated in Japan and the *se^del^* allele in South Asia [7,13]. On the other hand, the origin of the *se^del2^
*allele was unclear in some respects because the history of the Oceania population is complex. We first found this allele in a Samoan population (Polynesians, also called Remote Oceanians) and later in several New Guinean populations (Melanesians, also called Near Oceanians) [7,11]. Previous studies suggested that Polynesians have genetically interbred with East Asians (Taiwanese) and indigenous Melanesians many times [27,28,29]. The *se^del2^* allele has been found in Papuan- and Austronesian-speaking peoples of Irian Jaya (West Papua, Indonesia) and in mixed highland and coastal Papua New Guineans, but not in New Guinean highlanders (Dani peoples) [11]. Polynesians had a high frequency of c.385A>T and c.571C>T, which are common in Taiwanese populations, while these two alleles were rare or almost absent in Melanesians with or without the *se^del2^
*allele [7,8,10,30]. In addition, the *se^del2^
*allele does not to appear to be present in peoples from East and Southeast Asia, such as the Taiwanese and Indonesians [9,10]. Therefore, the *se^del2^
*allele is thought to have originated in an Oceanian population, rather than in an Asian population. However, it is still unclear whether the *se^del2^
*allele originated from Near or Remote Oceania. It would be desirable to conduct a large-scale study of the distribution of the *se^del2^
*allele in various populations in Oceanian and neighboring areas in order to determine the distribution and origin of this allele. 

In addition, the c.375A>G in *FUT2* is a very interesting SNP. This is because in gorillas and orangutans, the nucleotide at position 375 in *FUT2* is G, while in chimpanzees and humans, it is A. Therefore, it is assumed that the substitution from G to A occurred in the common ancestor of chimpanzees and humans, and that the re-substitution from A to G occurred once again in humans. Furthermore, c.375A>G appears to be relatively unique to Papuan-speaking Melanesian populations, although it was also present Austronesian-speaking Melanesian populations and Polynesian populations with a relatively low frequency [11]. In addition, c.375A>G and c.571C>T have recently been found in two late Neanderthals who lived in Russia and Croatia about 500,000 to 800,000 years ago [31]. However, these two Neanderthals had c.375A>G accompanied by c.571C>T, whereas in the Samoan and Taiwanese c.571C>T, it was not accompanied by c.375A>G [7,8,9,10,31]. Furthermore, one late Neanderthal in Altai and one Denisovan without c.571C>T also had c.375A>G as a homozygous state. One Denisovan had a c.400G>A as a heterozygous state. However, as with c.571C>T, the Denisovan c.400G>A was accompanied by c375A>G, but the Oceanian c.400G>A was not accompanied by c375A>G [11,31]. c.400G>A and c.375A>G were found at relatively high frequencies in Melanesians but were quite rare or not found in other populations [11]. 

Previous studies have suggested that a complex pattern of natural selection, such as balancing selection or positive selection in different populations, might act on the *FUT2* locus [6]. The selective force is presumed to be several pathogens. In fact, non-secretors have been reported to be highly resistant to noroviruses and rotaviruses, and more recently, to COVID-19 infection, albeit weakly [32,33,34]. In addition, recent studies have suggested that secretor status affects susceptibility to a variety of clinical conditions, including some infectious diseases, inflammatory bowel disease, and reduced plasma vitamin B_12_ levels [35,36,37,38,39,40,41,42]. Therefore, to understand the complex evolutionary history of the *FUT2* locus, it is very important to know the origin of these alleles and their relationship to the late Neanderthals, Denisovans, and Melanesians. From these perspectives, the present method is useful for estimating the origin of these variations by knowing their distribution.

## 5. Conclusions

In this study, we developed a method for the simultaneous detection of c.375A>G, c.385A>T, and c.571C>T, as well as the *se^del2^
*allele, via FMCA and an endpoint genotyping assay in a single tube. The present FMCA and endpoint genotyping method appeared to be reliable high-throughput methods for detecting c.375A>G, c.385A>T, and c.571C>T, as well as the *se^del2^
*allele, and for estimating secretor status in Oceanian populations. In addition, this method may be useful for examining the distribution and origin of these alleles.

## Figures and Tables

**Figure 1 diagnostics-13-02022-f001:**
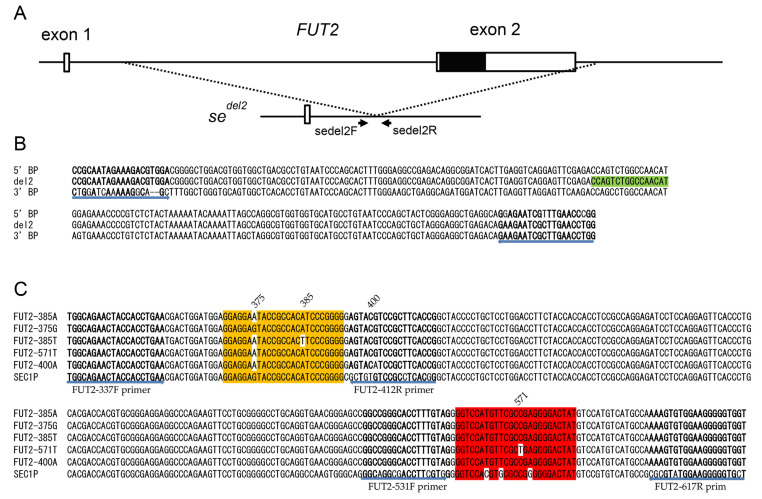
(**A**) Genetic structures of *FUT2* and *se^del2^*. Two exons are indicated by white boxes, and the protein coding region is indicated by a black box. Positions of PCR primers for duplex real-time PCR are shown by arrows. (**B**) Alignment of DNA sequences of amplified regions of 231 bp surrounding the *se^del2^* breakpoint. DNA sequences of the 5’ breakpoint (5’ BP), 3’ breakpoint (3’ BP), and junction region (del2) are indicated. Primer sequences are shown in bold, and the hydrolysis probe sequence (sedel2-FAM) is boxed in light green. The positions and directions of the primers are indicated by arrows below the alignment. (**C**) Alignment of DNA sequences of five *FUT2* alleles (FUT2-385A, -375G, -385T, -571T, and -400A) and corresponding regions of SEC1P are indicated. Primer sequences are shown in bold, the 375-probe sequence is highlighted in orange, and the 571-probe sequence is highlighted in red. Mismatched nucleotides in each probe are shown without highlighting. The positions and directions of the primers are indicated by arrows below the alignment.

**Figure 2 diagnostics-13-02022-f002:**
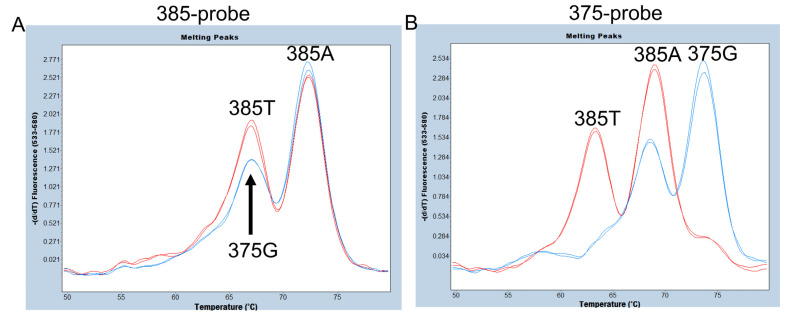
Melting peak profiles of the 385-probe (**A**) and of the 375-probe (**B**) for heterozygotes of c.385A>T (385A/T) (red) and c.375A>G (375A/G) (blue). Duplicated results using two samples are shown. (**A**) The peak height of 375G seems to be slightly lower than that of 385T, but accurate separation is considered difficult using a 385-probe. (**B**) The three melting peaks corresponding to 375G, 385A, and 385T alleles were completely separated using the 375-probe.

**Figure 3 diagnostics-13-02022-f003:**
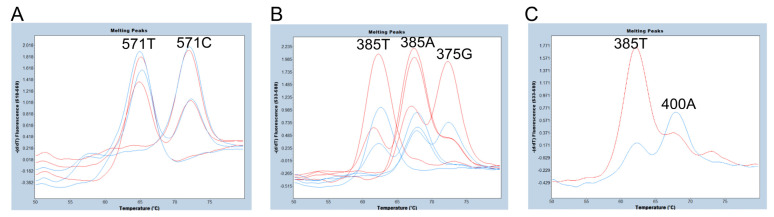
Melting peak profiles of the 571-probe (**A**) and the 375-probe (**B**,**C**) for selected subjects using one or two amplicons. (**A**) Melting peak profiles of the 571-probe obtained from 571C/C, 571C/T, and 571T/*se^del2^* subjects with two amplicons (red) and those with one amplicon (blue). The peak heights obtained with two amplicons and those with one amplicon are almost equivalent. (**B**) Melting peak profiles of the 375-probe obtained from 385A/A, 385A/T, 385T/T, and 385A/375G subjects with two amplicons (red) and those with one amplicon (blue). The peak heights obtained with two amplicons are higher than those with one amplicon. (**C**) Melting peak profiles of the 375-probe obtained from one heterozygote of 400G/A with two amplicons (red) and one amplicon (blue). The peak height corresponding to 385A (400A allele) is quite low in this subject when using two amplicons, while the melting peak profile is similar to that of the 385A/T heterozygote when using one amplicon.

**Figure 4 diagnostics-13-02022-f004:**
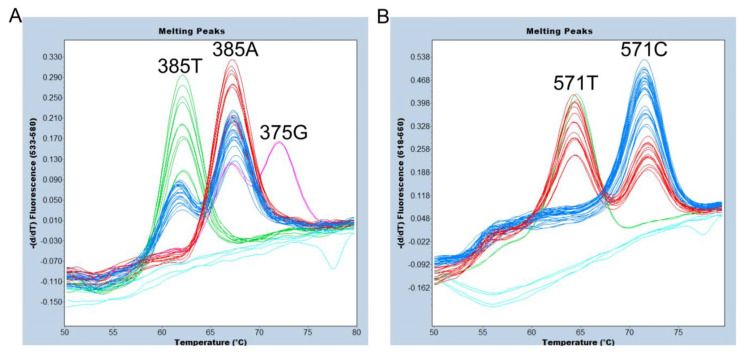
Melting peak profiles of the 375-probe (**A**) and of the 571-probe (**B**) for 24 Samoans. Duplicated results are shown. (**A**) The subjects with 385A/A (red), A/T (blue), and T/T (green) genotypes were completely separated. In addition, one 375G/385A (pink) heterozygote was clearly separated. Negative controls and one *se^del2^* homozygote are indicated by light blue. (**B**) The subjects with 571C/C (blue), C/T (red), and T/T (green) genotypes were completely separated. Negative controls and one *se^del2^* homozygote are indicated by light blue.

**Figure 5 diagnostics-13-02022-f005:**
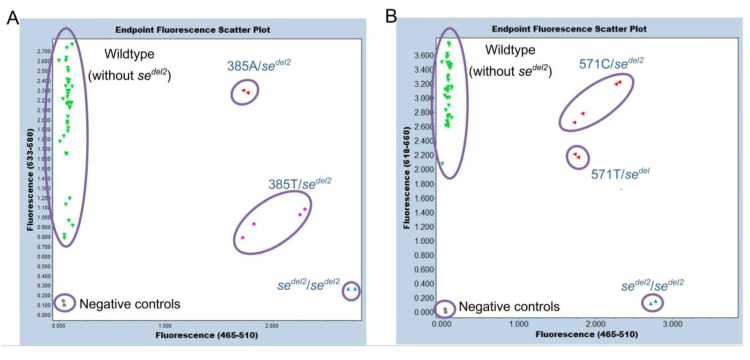
Endpoint genotyping of *se^del2^*. Duplicated results for genomic DNA from 24 Samoans are shown. (**A**) The results of a dual-color scatter plot of fluorescence signals by 465–510 nm/533–580 nm. The subjects of wild types—without *se^del2^* (green), *se^del2^* heterozygotes—385A/- (red), *se^del2^* heterozygotes—385T/- (pink), and *se^del2^* homozygotes (blue) are separated. Negative controls are indicated by gray. (**B**) The results of a dual-color scatter plot of fluorescence signals by 465–510 nm/618–660 nm. The subjects of wild types—without *se^del2^* (green), *se^del2^* heterozygotes—both 571C/- and 571T/- (red), and *se^del2^* homozygotes (blue) are separated. Negative controls are indicated by gray.

**Table 1 diagnostics-13-02022-t001:** Primers and probes for detection of c.375A>G, c.385A>T, and c.571C>T of *FUT2*.

Primer Sequences	Position of *FUT2*	Position of *SEC1P*	Differences with *SEC1P*	Amplicon Length
Detection of 375A>G, 385A>T				
FUT2-337F: 5′-TGGCAGAACTACCACCTGAA-3′	337–356 bp	379–398 bp	0	76 bp
FUT2-412R: 5′-CGGTGAAGCGGACGTACT-3′	395–412 bp	437–454 bp	6	
385-probe: 5′-HEX GGAGGAATACCGCCACATCCCGGGG-BHQ1-3′	369–393 bp	411–435 bp	1	
375-probe: 5′-HEX GGAGGAGTACCGCCACATCCCGGGG-BHQ1-3′	369–393 bp	411–435 bp	0	
Detection of 571T>C	539–556 bp	581–598 bp	5	
FUT2-531F: 5′-GGCCGGGCACCTTTGTAG-3′	598–617 bp	640–659 bp	5	79 bp
FUT2-617R: 5′-ACCACCCCCTTCCACACTTT-3′	558–582 bp	600–624 bp	3	
571C-probe: 5′-Cy5-GGTCCATGTTCGCCGAGGGGACTAT-BHQ2-3′	539–556 bp	581–598 bp	5	
Detection of 375A>G, 385A>T, 571T>C				
FUT2-337F: 5′-TGGCAGAACTACCACCTGAA-3′	337–356 bp	379–398 bp	0	281 bp
FUT2-617R: 5′-ACCACCCCCTTCCACACTTT-3′	598–617 bp	640–659 bp	5	
375-probe and 571-probe				

“Differences with *SEC1P*” indicates the number of base differences in each of primer or probe with corresponding *SEC1P* sequences. FUT2-337F primer and FUT2-412 R primer: same primers as previously used for 385A>T detection via unlabeled probe HRM analysis [23]. 385-probe: same probe (TaqMan probe for c.385A>T, labeled with 5′-HEX and 3′-BHQ1) as previously used for detection of c.385A>T [24]. 375-probe: same probe (formerly named SEC1P-FUT2 probe) as previously used for simultaneous detection of *se^fus^
*and c.385A>T [13]. BHQ: black hole quencher.

## Data Availability

The data presented in this study are available upon request from the corresponding author.

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
