# Peer review of "Detection of c.375A>G, c.385A>T, c.571C>T, and sedel2 of FUT2 via Real-Time PCR in a Single Tube"

_diagnostics, 2023, doi:10.3390/diagnostics13122022_

Round 1

Reviewer 1 Report

In the introduction, there is no explanation about the HEX-label probe. However, it is chosen as a probe in this study.

It is essential to state the aim of this study in the introduction.

The discussion should focus on comparing the advantages and disadvantages of this new method with the previous publication because the title is related to proving the new methods. 

It is important also to state whether there is a limitation in this study and what the author suggests for future study.

Author Response

To Reviewer 1

Thank you very much for your evaluation of our manuscript. Following your suggestions, we revised our manuscript. Changes were made as follows:

Q1: In the introduction, there is no explanation about the HEX-label probe. However, it is chosen as a probe in this study.

Our reply: Thank you for your comment. Following your suggestion, we added explanation about the HEX-label probe in introduction. “Since HEX is the most widely used fluorophore that can be used in the same filters as VIC, HEX probes were used instead of VIC.” (lines 76-77)  

Q2: It is essential to state the aim of this study in the introduction.

Our reply: Thank you for your comment. Following your suggestion, we added the aim of this study in introduction.” In this study, we attempted to develop a multiplex FMCA for the purpose of developing a genetic analysis method that enables simple and rapid estimation of the secretor status of the Oceanians.” (lines 74-76)

Q3: The discussion should focus on comparing the advantages and disadvantages of this new method with the previous publication because the title is related to proving the new methods.

Our reply: Thank you for your comment. Following your suggestion, we discussed about advantages and disadvantages of this new method with the previous publication.” Direct Sanger sequencing of PCR products……the distribution of variations” (lines 299-311)

Q4: It is important also to state whether there is a limitation in this study and what the author suggests for future study.

Our reply: Thank you for your helpful suggestion. Following your suggestion, we added the limitations of the present method. “ Limitations of this method are that it cannot detect se alleles other than Sew, se571, and sedel2, and it is not very useful for FUT2 polymorphism analysis outside of Oceania.” (lines 311-313)

Reviewer 2 Report

In this study, the authors presented a research entitled as, “Detection of c.375A>G, c.385A>T, c.571C>T, and sedel2 of FUT2 2 by real-time PCR in a single tube”. The authors present a method based on the real-time PCR for detection three SNPs of FUT2 gene, including c.375A>G, c.385A>T, and 10 c.571C>T, and sedel2. It is an interesting method for determination of the nucleotide variants. However, there are some problems in the text. The comments are as shown in the following.

1.          The description of the materials used for detection of FUT2 gene focused on the detection of c.375A>G, c.385A>T, and 10 c.571C>T, but the detection of sedel2 was ignored and less mentioned. Therefore, the authors might described more about the detection of sedel2. For example, how about the sequencing and fluorescent labelling of the probe, and the primers.

2.          In all figures, the 375A could not been seen and checked in the use of 385-probe and 375-probe. Please explain?

3.          The authors said the results was confirmed by Sanger sequencing in line 241. The authors could also show the data of Sanger sequencing to demonstrate these sequences were correct.

4.          In legend of Figure 2, the sentence was as following: Melting peak profiles of the 385-probe (A) and of the 375-probe (B) for heterozygote of 185 c.385A>T (385A/T) (red) and c.375A>G (375G/385A) (blue). If this sample of c.375A>G “(375G/385A)” could be considered as a heterozygote?? Please check and correct that.  

The quality of English is moderate, and could let the readers know what the authors want to express.    

Author Response

To Reviewer 2

Thank you very much for your evaluation of our manuscript. Following your suggestions, we revised our manuscript. Changes were made as follows:

Q1: The description of the materials used for detection of FUT2 gene focused on the detection of c.375A>G, c.385A>T, and c.571C>T, but the detection of sedel2 was ignored and less mentioned. Therefore, the authors might be described more about the detection of sedel2. For example, how about the sequencing and fluorescent labelling of the probe, and the primers.

Our reply: Thank you for your helpful suggestion. Following your suggestion, we revised Figure 1 to explain sedel2. (Figure 1A and B and the legend for figure)

Q2:  In all figures, the 375A could not been seen and checked in the use of 385-probe and 375-probe. Please explain?

Our reply: I'm sorry to have confused you. As explained for manuscript, only one subject is heterozygous for 375A/G and the remaining 23 Samoan samples are all homozygous for 375A (375A/A). The 375A allele is further segregated into 385A and 385T alleles at c.385A>T; the nucleotide at position 385 of the 375A allele is an A, indicating that the 375A/G subjects are heterozygous for the 375G and 385A alleles. We have added this explanation. (lines 180-182)

Q3: The authors said the results was confirmed by Sanger sequencing in line 241. The authors could also show the data of Sanger sequencing to demonstrate these sequences were correct. We added the representative Sanger sequencing results for wild-type homozygote, heterozygote and variant-type homozygote of c.385A>T and c.571C>T, in supplementary Figure 1. (lines 256-258 and supplementary Figure 1)

Our reply:

Q4: In legend of Figure 2, the sentence was as following: Melting peak profiles of the 385-probe (A) and of the 375-probe (B) for heterozygote of c.385A>T (385A/T) (red) and c.375A>G (375G/385A) (blue). If this sample of c.375A>G “(375G/385A)” could be considered as a heterozygote?? Please check and correct that. 

Our reply: As mentioned above, only one subject is heterozygous for 375A/G and the remaining 23 Samoan samples are all homozygous for 375A (375A/A). However, we changed “(375A/385A)” to “(375A/G) in Figure 2 legend. (line 200)

Reviewer 3 Report

This study aims at detecting genetic variations of FUT2 (encoding Se enzyme involved in secretor status of ABH(O) blood group antigens) by real-time PCR in a single tube. Results indicated successful simultaneous detection of FUT2 variations in the Oceanian population using HEX- and Cy5-labeled probes instead of VIC-labeled probes. Using endpoint genotyping analysis and a multiplex approach, the authors suggested a time and cost-efficient detection of three SNPs and zygosity determination. The manuscript is well written; the results are correctly analyzed and support the conclusions that are concisely presented in the discussion”.

Author Response

Thank you very much for your evaluation of our manuscript.

Reviewer 4 Report

In this manuscript, the author employs a previously established methodology, pioneered by the author themselves, to investigate the potential benefits of a newly identified gene. While the method utilized is undoubtedly intriguing, it is worth noting that it has already been disclosed in previous publications. Consequently, the author's primary contribution lies in the subsequent assessment of the real-time PCR technique on these novel genes. However, the current level of evaluation falls short of meeting the rigorous standards expected for publication in a distinguished journal like Diagnostics

Author Response

(The authors gave the same response as above.)

Round 2

Reviewer 1 Report

None